# Wave-resolving Voronoi model of the Rouse number for sediment entrainment

**Johannes Lawen**

Hamburg University of Technology, Am Schwarzenberg-Campus 4 (C), 21073 Hamburg, Germany

**Correspondence:** Johannes Lawen (jl@environment.report)

**Abstract.** `CEI` To integrate wave and sediment transport modeling, a computationally extensive wave-resolving Voronoi-mesh-based simulation has been developed to improve upon heretofore separate sediment and spectral wave modeling. Orbital wave motion-dependent sediment transport and fine structures of the dynamic Rouse number distribution across the seabed were brought into focus. The entirely parallelized wave-resolving hydrodynamic model is demonstrated for nearshore beach waters adjacent to artificial islands in Doha Bay. The nested model was validated with tidal time series for three locations and two seasons.

## 1 Introduction

Wave-resolving hydrodynamics, similar to direct numerical simulations (DNSs) of turbulent eddy mixing, requires resolutions that are usually impossible or impracticable to compute within even logarithmic orders in the teraflops (floating-point operations per second) range. Therefore, waves and sediment transport have heretofore been simulated separately (Hsu et al., 2005; Anastasiou and Sylaios, 2013; Yu et al., 2018), entailing a principle limit in rigor. This work advances the integration of both wave and sediment transport modeling into a single direct wave simulation by exploiting the filtering of shallow orbital wave motion. That is, wave motion not in contact with the seabed does not need to be resolved to depict entrainment. Waves modeled here thus do not encompass the entire wavelength spectrum, which would also include short waves that barely perturb shear forcing on the seabed. Therefore, with respect to bottom shear, wave-resolving computation as an alternative to spectral models (La Forgia et al., 2023) can be achieved well before full-coastal DNSs. This approach might only be valid for certain wave regimes and climates and may be less suited for others that exhibit waves too short to resolve but long enough to be in contact with the seabed.

If conditions are calm, with wind waves exhibiting wavelengths much smaller than twice the water depth, then the perturbation of near-bottom tidal currents due to orbital wave motion remains negligible. If conditions are sufficiently agitated, such that wavelengths reach the order of magnitude of the water depth in size, then wave orbital motion may considerably influence bottom currents. Simulating waves and sediment transport in an integrated model that is capable of resolving waves enhances the rigor in depicting these shear forces.

The sections below contain the wave-resolving simulation of the Rouse number distribution to resolve small scales in the balance of sediment entrainment and deposition. The Voronoi model developed here is suitable for wave resolution, as the variable number of edges per finite volume reduces wave fronts on acute triangle angles. In terms of numerical diffusion (Holleman et al., 2013), Voronoi meshes exhibit a reduction compared to Delaunay meshes (Chan et al., 2018). Analytical verification of the model made possible by dynamic domain contractions has been documented separately (Lawen, 2023). Earlier triangle-mesh-based versions of the model have been in use for a decade (Lawen et al., 2013, 2014) for studies of reactive transport. Several reasons might have initially supported the choice of unstructured triangle meshes (Lawen et al., 2014), which followed prolific (Falcieri et al., 2014; Ricchi et al., 2017) models based on structured meshes (Lawen et al., 2010; Ladant et al., 2024) vis-á-vis Voronoi meshes: for example, cells in the latter mesh type are formed by a vary-

ing number of edges. Without sparse matrix storage mode, the sizes of arrays for vertices and faces are, thus, determined by the Voronoi cell with the most faces. Meanwhile, triangle cells always have just three horizontal faces (Lawen et al., 2013, 2014; Cousins et al., 2013; Tadesse and Fröhle, 2020), yielding compact arrays for early cache-fitting simulations of multiple bus-snooping schemes and, thus, cache-coherent cores or CPUs. Voronoi meshing has lately also been applied to oceanography, with work (Herzfeld et al., 2020; Ringler et al., 2013) mentioning certain stability concerns vs. Delaunay-mesh-borne models. That is, an algorithm that might be stable on a Delaunay mesh might not necessarily be stable on a Voronoi mesh. The need for the development of ocean models based on Voronoi meshes has been put forth with dedicated emphasis (Ringler et al., 2013): "all 23 global ocean models used in the Intergovernmental Panel on Climate Change (IPCC) 4th Assessment Report (Intergovernmental Panel on Climate Change, 2007) were based on structured, conforming quadrilateral meshes (see Chap. 8, p. 597, of Randall and Bony, 2007). Our view is that the global ocean modeling community benefits from having a diversity of numerical approaches. While this diversification is well underway with respect to the modeling of the vertical coordinate (Hallberg, 1997; Bleck, 2002), progress in developing new methods for modeling the horizontal structure of the global ocean on climate-change timescales has lagged behind".

The ADCIRC (Pringle et al., 2021; Kerr et al., 2013; Shashank et al., 2021; Szpilka et al., 2022), FESOM (Wang et al., 2014), Fluidity-ICOM (Kimura et al., 2013), FVCOM (Yu et al., 2017), ICON (Mehlmann and Korn, 2021; Logemann et al., 2021; Linardakis et al., 2022), Mike 3 Wave FM (Kaergaard et al., 2019), SCHISM (Lynett et al., 2017, formerly SELFE), SLIM (Dobbelaere et al., 2024; Sterckx et al., 2023; Vincent et al., 2022), SUNTAS (Fringer et al., 2006; Masunaga et al., 2023), Thetis (Scott et al., 2023; Wallwork et al., 2024; Mawson et al., 2022), and UNTRIM (Casulli, 1999; Mahavadi et al., 2024) models simulate triangular and/or quadrilateral meshes. The D-Flow FM (Frölke, 2016) and E3SM (Feng et al., 2022; Leung et al., 2020; Petersen et al., 2019; Golaz et al., 2019) models, including configurations such as the MPAS (Lilly et al., 2023, 2025; Pal et al., 2023), also process pentagons and hexagons. The Voronoi-mesh-borne model provided here further contributes to the requested diversity.

Model validation was performed with five correlations of simulated surface elevations, with time series from a tidal survey of three locations and two seasons. In terms of the hydrodynamic components examined in the validation, reliable validation often relies on time series for surface elevations, as conceded in a number of works (Hsu et al., 1999; Blumberg, 1977; Oey et al., 1985; Park and Kuo, 1993; Muin and Spaulding, 1996) that found that the simulated quantity exhibits the best correlation with survey measurements. This follows from the magnitudes of the hydrostatic and momentum terms and has been observed in general, including in recent works (Lawen et al., 2013, 2014; Yu et al., 2017). That is, Earth's gravity levels water tables such that tidal constituents are usually rather well-behaved quantities in comparison to velocity components. The bottom drag was calibrated to vertical current profiles to obtain reliable values for bottom roughness.

## 2 Method

The Cauchy partial differential equation (PDE) adds the depiction of stresses to the Euler momentum PDE. In the Navier–Stokes PDE, the stresses of the Cauchy PDE are specified for Newtonian fluids, that is, molecular momentum dissipation is proportional to the fluid's shear rate. Reynolds-averaged Navier–Stokes (RANS) simulations and large-eddy simulations (LES) harness momentum transport by utilizing the diffusive term in the Navier–Stokes PDE. Approximations and configurations of the latter for coastal ocean domains are known as shallow-water equations (SWEs) or primitive equations, primitive in the sense of the fundamental governing function. The model solves, in conjunction with the continuity PDE (Eq. 10), the incompressible Navier–Stokes PDE (Eq. 1) configured for surface flow, which accounts in the control volume $V$ (m$^3$)

$$\frac{\partial (\rho \, \boldsymbol{u} \, V)}{\partial t} + \sum \frac{\partial (u_i \rho_i \, \boldsymbol{u} \, V)}{\partial x} = \boldsymbol{F} + \nabla \cdot \begin{pmatrix} \tau_{xx} \tau_{xy} \tau_{xz} \\ \tau_{yx} \tau_{yy} \tau_{yz} \\ \tau_{zx} \tau_{zy} \tau_{zz} \end{pmatrix} \quad (1)$$

for the component velocities $\boldsymbol{u} = [u \; v \; w]$ (m s$^{-1}$), with the force $\boldsymbol{F}$ (kg m s$^{-2}$) due to the hydrostatic pressure gradient and Coriolis acceleration, as detailed in Sect. 2.2 and 2.3. The stress tensor $\tau_{ij}$ (kg m s$^{-2}$) is configured for an incompressible Newtonian fluid and horizontally isotropic viscosity in Sect. 2.4. Finite volume approximations are demonstrated for all terms in the following dedicated subsections: Sect. 2.1 for continuity, Sect. 2.2 for advection and hydrostatic pressure, Sect. 2.3 for Coriolis acceleration, Sect. 2.4 for viscous stress, Sect. 2.5 for the Smagorinsky coefficient, Sect. 2.6 for hydrodynamic boundary conditions, Sect. 2.7 for sediment transport, and Sect. 57 for erosion. These subsections also incorporate derivations of the respective terms.

If component velocities are uniform throughout a finite volume, then the latter is termed convective: quantities are uniformly "convected" throughout a cell. If component velocities are nonuniform throughout a finite volume, then an algorithm is termed conservative if the same quantities exit and enter adjacent finite volumes through faces. That is, constituents are conserved and not lost throughout the domain. Variable velocities within one finite volume, that is, the conservative case, correspond to transport velocities remaining in the PDE's derivative. Transport velocities can be arranged outside the PDE's derivative, corresponding to uniform velocity components in a finite volume if

the continuity PDE is inserted into the PDE of the quantities concerned. The PDE system is then termed convective. A uniform velocity is obviously more suitable to warrant constituent emission from a finite volume. Therefore, conservative algorithms are less likely to ascertain the stability of the simulation, particularly if errors are repetitively amplified in circulation (Lawen et al., 2014). Both cases are here derived via the constituent balance of a finite volume. While considering a convective case of quantity transport, the conservative case is still used to derive the algorithm for the continuity PDE. Summation indices range over $x$, $y$, and $z$ for derivatives and for finite volume terms, over the faces of the Voronoi polygon. A control volume balance of quantity flows $j_i$ along $x_i$ (m) for dimension $i$ and quantity $f$ yields

$$\frac{\partial f}{\partial t} = -\sum \frac{\partial j_i}{\partial x_i} \tag{2}$$

for $n$ spatial dimensions. That is, for advection with velocity components $u$, $v$, and $w$ (m s$^{-1}$),

$$\frac{\partial f}{\partial t} = -\sum \frac{\partial (u_i f)}{\partial x_i}. \tag{3}$$

The finite volume approximations utilized to convert the partial differential equations for momentum, continuity, and scalar transport into finite volume equations are listed in Table 1 below.

Most of the upwind and central difference algorithms listed have been used before in a similar manner in the 3D Simulation for Marine and Atmospheric Reactive Transport (3D SMART) (Lawen et al., 2013, 2014) for other transport quantities, such as height $h$ (m), temperature $T$ (°C), salinity $S$ (PSU), and concentration $c$ (kg m$^{-3}$), and on triangle meshes instead of Voronoi meshes.

These approximations of derivatives are expressed in Table 2 in areas $A$ (m$^2$) and edges $e$ (m) alongside an evaluation of derivatives based on the total differential. $\mathbb{1}_{>0}(q)$ denotes the indicator function that evaluates whether the volume flow $q_i$ (m$^3$ s$^{-1}$) through face $i$ into the finite volume fulfills a particular logical condition, such as for influx and efflux or whether the flow is entering or exiting, to facilitate upwinding. $n_i(f)$ denotes here a quantity value at the centroid of a face shared with neighbor $i$ in the neighbor list.

The Wavedyne, the new Voronoi-mesh-borne version of the 3D SMART, features a gradient computation based on the total derivative: a polygon's centroid and the centroids of two neighboring cells $\beta$ and $\gamma$ constitute a triangle. That is, the total derivative is denoted for one of two edges of a triangle, formed between the centroid of a particular cell and the centroids of two adjacent cells, $\beta$ and $\gamma$, if $\beta$ is likewise adjacent to $\gamma$,

$$\delta f_1 = \frac{\partial f_1}{\partial x} \delta x_1 + \frac{\partial f_1}{\partial y} \delta y_1, \tag{4}$$

with particular edge components $\delta x_i$, $\delta y_i$, and $\delta f_i$ for edge $i$ but with a common $\partial f/\partial x$ and $\partial f/\partial y$ throughout the triangle,

yielding

$$\frac{\partial f}{\partial x} = \left( \delta f_1 - \frac{\partial f}{\partial y} \delta y_1 \right) / \delta x_1. \tag{5}$$

Likewise, the total derivative can be denoted for another edge, and the gradient from the left-hand side (LHS) of Eq. (5) is inserted:

$$\delta f_2 = \left( \delta f_1 - \frac{\partial f}{\partial y} \delta y_1 \right) \frac{\delta x_2}{\delta x_1} + \frac{\partial f}{\partial y} \delta y_2, \tag{6}$$

which may be resolved for the complementary gradient

$$\frac{\partial f}{\partial y} = \frac{\delta f_2 - \delta f_1 \delta x_2 / \delta x_1}{\delta y_2 - \delta y_1 \delta x_2 / \delta x_1}. \tag{7}$$

Assembling the gradient for asymmetrical Voronoi cells out of triangles requires, furthermore, weighting factors. In Table 2, binary $\alpha$ takes a logical functionality, carrying the value 0 or 1, and is calculated before the simulation to select edges 1 and 2 such that division by small numbers is avoided, enhancing accuracy. Likewise, whereas the total derivative of $\partial f/\partial y$ in Table 2 is calculated before $\partial f/\partial x$, the calculation is also conducted in inverted order to provide a substitute in case of division by zero.

Arrays for $\alpha$, $\beta$, and $\gamma$ are computed once and for all before the simulation, as these arrays depend only on the mesh geometry. Nevertheless, the computational costs of the procedure cannot be said to be negligible, as they lead to a doubling of the run time vis-á-vis the central difference approximation listed in Tables 1 and 2. Algorithm validation has been conducted using a method of manufactured solutions (MMS), which has been submitted separately for publication (Lawen, 2024a). The MMS was realized by oscillating the seabed to match the flow field to an analytical solution.

## 2.1 Continuity

The continuity PDE is obtained by specifying the transport PDE (Eq. 2) for mass. The conservative form is converted into the convective form by applying the product rule for derivatives in the equation for mass continuity.

$$\frac{\partial m}{\partial t} = -\sum \frac{\partial \dot{m}_i}{\partial x_i} \tag{8}$$

With $\partial m = \partial (\rho V)$ (kg m$^{-3}$ m$^3$), this is rendered as

$$\frac{\partial (\rho V)}{\partial t} = -\sum \frac{\partial (u_i \rho V)}{\partial x_i}, \tag{9}$$

dividing $V = ph$ by the polygon area $p$ in a convective form, that is, after application of the product rule

$$\frac{\partial (\rho h)}{\partial t} = -\sum u_i \frac{\partial (\rho h)}{\partial x_i} - \rho h \sum \frac{\partial u_i}{\partial x_i}. \tag{10}$$

On the right-hand side (RHS) of the approximation below, the first term exhibits the convective form for the quantity $h$,

**Table 1.** Term approximation.

| Term | Convective upwind | Conservative central difference | Central difference | Diffusive central difference | Total derivative |
|---|---|---|---|---|---|
| Advect $u, v, w, h\rho, T, S, c$ | $\nabla$ | | | | |
| Advect $q$ | | $\nabla$ | | | |
| Hydrostatic pressure | | | $\frac{\partial}{\partial x_i}$ | | $\frac{\partial}{\partial x_i}$ |
| Viscous diffusion | | | | $\nabla^2$ | |
| Eddy shear rates | | | $\frac{\partial}{\partial x_i}$ | | $\frac{\partial}{\partial x_i}$ |
| $\nabla$ in eddy viscosity's $\nabla \cdot \tau$ | | | $\frac{\partial}{\partial x_i}$ | | $\frac{\partial}{\partial x_i}$ |
| Smagorinsky model | | | $\frac{\partial}{\partial x_i}$ | | $\frac{\partial}{\partial x_i}$ |

**Table 2.** Finite volume approximations.

| | Integrated quantity flux approximation |
|---|---|
| $f'\Big|_{\substack{\text{convective} \\ \text{upwind}}}$ | $A^{-1}\sum_i \left[ q_i \left( n_i(f)\mathbb{1}_{>0}(q_i) + f_i\mathbb{1}_{\leq 0}(q_i) \right) \right]$ |
| $f'\Big|_{\substack{\text{conservative} \\ \text{central}}}$ | $(2A)^{-1}\sum_i \left[ q_i f_i + n_i(q)n_i(f) \right]$ |
| $f'\Big|_{\text{central}}$ | $(2A)^{-1}\sum_i \left[ \mathrm{proj}_{\perp r}(\boldsymbol{e}_i)\left(n_i(f) - f\right) \right]$ |
| $f''\Big|_{\text{central}}$ | $A^{-1}\sum_i \left[ |\boldsymbol{e}_i|\left(n_i(f) - f\right) \right]$ |
| $f_y\Big|_{\substack{\text{total} \\ \text{derivative}}}$ | $\frac{\alpha(f_\beta - f) + 0^\alpha(f_\gamma - f) - (\alpha(f_\gamma - f) + 0^\alpha(f_\beta - f))\delta x_2/\delta x_1}{\delta y_2 - \delta y_1 \delta x_2/\delta x_1}$ |
| $f_x\Big|_{\substack{\text{total} \\ \text{derivative}}}$ | $\left( \alpha\left(f_\gamma - f\right) - 0^\alpha\left(f_\beta - f\right) - \delta y_1 \partial f/\partial y \right)/\delta x_1$ |

and the second term exhibits the conservative form for a constant quantity equal to 1. Two corresponding finite volume approximations, for the convective and conservative cases, respectively, can be inserted from Table 2. The first term is approximated with upwinding and the second term with a central difference approximation. $^{+\delta t}$ denotes a quantity at the subsequent time level. Past triangle mesh-versions of the 3D SMART species transport (Lawen et al., 2013, 2014) also featured semi-implicit matrix reordering algorithms. However, these attained only a tripling of time steps at the expense of flops for the reordering, rendering the net computational gain questionable.

$$\left(h^{+\delta_t}\rho^{+\delta_t} - h\rho\right)\frac{p}{\delta_t}$$

$$= \sum \overbrace{\left[ \frac{q_i}{h_i}\left( n(h_i\rho_i)\mathbb{1}_{>0}(q_i) + h_i\rho_i\mathbb{1}_{\leq 0}(q_i) \right) \right]}^{\text{upwind}}$$

$$+ \frac{h\rho}{2} \sum \overbrace{\left[ \frac{q_i}{h_i} + \frac{n(q_i)}{n(h_i)} \right]}^{\text{central difference}} \tag{11}$$

Here, $q_i$ is the volume flow through face $i$ based on the component velocities at the cell's centroid. Meanwhile, $n(q_i)$ is the volume flow based on the component velocities of the neighbor at face $i$. The latter is included to approximate the volume flow at the face between two cells. The total horizontal flow through cell faces, the summed-up component volume flows, are the products of component velocities and the perpendicular edge components. That is,

$$\frac{q_i}{h_i} = \mathrm{proj}_{\perp x}(\boldsymbol{e}_i)u - \mathrm{proj}_{\perp y}(\boldsymbol{e}_i)v. \tag{12}$$

## 2.2 Convective material derivative and hydrostatic pressure

In addition to the time derivative, the material derivative also contains advective momentum transport in the Euler momentum, Cauchy, Navier–Stokes, shallow-water, and primitive equations. Inserting momentum into the PDE for quantity advection yields

$$\frac{\partial (\boldsymbol{u}\rho V)}{\partial t} = -\sum \frac{\partial (u_i \rho \boldsymbol{u} V)}{\partial x_i}. \tag{13}$$

Applying the product rule to the LHS and RHS yields

$$\boldsymbol{u}\frac{\partial (\rho V)}{\partial t} + \rho V \frac{\partial \boldsymbol{u}}{\partial t} = -\sum \left[ u_i \rho V \frac{\partial \boldsymbol{u}}{\partial x_i} + \boldsymbol{u}\frac{\partial (u_i \rho V)}{\partial x_i} \right]. \tag{14}$$

Inserting the conservative form of the continuity PDE into the LHS yields the opportunity to eliminate terms, returning only one term each for the LHS and RHS:

$$\rho V \frac{\partial \boldsymbol{u}}{\partial t} = -\rho V \sum \left[ u_i \frac{\partial \boldsymbol{u}}{\partial x_i} \right]. \tag{15}$$

The fluid from Eq. (13) can be denoted by the velocity in its spatially differential form:

$$\partial (\rho\, \boldsymbol{u}\, V)/\partial t = -\sum \frac{\partial (\rho_i\, \boldsymbol{u}\, V)}{\partial x}\frac{\mathrm{d}x_i}{\mathrm{d}t}, \tag{16}$$

which matches the form of the RHS of the total derivative $\mathrm{d}f = \sum \partial f/\partial s\, \mathrm{d}s$, with $\mathrm{d}s = [\mathrm{d}x_1 \ldots \mathrm{d}x_n,\ \mathrm{d}t]$ divided by the time increment. Hence,

$$\frac{\mathrm{d}(\rho\boldsymbol{u}\, V)}{\mathrm{d}t} = \frac{\partial (\rho\, \boldsymbol{u}\, V)}{\partial t} + \sum \frac{\partial (\rho_i\, \boldsymbol{u}\, V)}{\partial x}\frac{\mathrm{d}x_i}{\mathrm{d}t} \tag{17}$$

or, in consideration of $u_i = \mathrm{d}x_i/\mathrm{d}t$,

$$\frac{\mathrm{d}(\rho\boldsymbol{u}\, V)}{\mathrm{d}t} = \frac{\partial (\rho\, \boldsymbol{u}\, V)}{\partial t} + \sum \frac{\partial (u_i\, \boldsymbol{u}\, \rho_i\, V)}{\partial x}. \tag{18}$$

PDE and the finite volume equation (FVE) can, hence, be configured as an Euler equation by including forces. As Newton's second law holds for force $\boldsymbol{F}$ and momentum $m\boldsymbol{u}$,

$$\boldsymbol{F} = \mathrm{d}(m\boldsymbol{u})/\mathrm{d}t. \tag{19}$$

Given that $m\boldsymbol{u} = \rho\, V\boldsymbol{u}$, the net force in the LHS is obtained by summing up all forces $F_j$ in a free-body diagram.

$$\sum \boldsymbol{F}_j = \sum \frac{\partial (u_i m\boldsymbol{u})}{\partial x_i} + \frac{\partial m\boldsymbol{u}}{\partial t} \tag{20}$$

In terms of FVE approximation, a quantity balance for a regularly or irregularly shaped finite volume transported with an upwind approximation in volume flows $q_i$ into volume $V$ returns for quantity $f$

$$\frac{\partial (fV)}{\partial t} = \sum \left( q_i \left( n(f_i)\mathbb{1}_{>0}(q_i) + f_i \mathbb{1}_{\leq 0}(q_i) \right) \right). \tag{21}$$

$\mathbb{1}$ is the indicator function that denotes the logical condition of using quantity values of neighboring cells for faces $i$ where volume flows $q_i$ are positive, that is, during inflowing. The component velocity basis for the volume flows determines whether this FVE corresponds to the conservative or the convective case. The latter is the case if the centroid's component velocities are applied to all faces. Inserting momentum into the FVE (Eq. 21) for quantity advection yields

$$\frac{\partial (\boldsymbol{u}\rho V)}{\partial t} = \sum_i \left( q_i \left( \rho n(\boldsymbol{u}_i)\mathbb{1}_{>0}(q_i) + \rho \boldsymbol{u}_i \mathbb{1}_{\leq 0}(q_i) \right) \right). \tag{22}$$

If the force acting on surfaces $i$ of the irregular fluid parcel is pressure, then as per $F_i = (\delta_P)_i \mathrm{proj}_{\perp r}(A_i)$, with the pressure difference $\delta_P$ and the vector of orthogonal component areas $\mathrm{proj}_{\perp r}(A_i)$, the following holds:

$$\sum_i \left( (\delta_P)_i \mathrm{proj}_{\perp r}(A_i) \right) = \sum \frac{\partial (u_i p)}{\partial x_i} + \frac{\partial p}{\partial t}. \tag{23}$$

Due to a difference in surface height $\delta_h/2$ between adjacent centroids and the edge of the considered cell, it follows that

$$g\sum_i \left[ \left( \rho \frac{\delta_h}{2} \right)_i \mathrm{proj}_{\perp r}(A_i) \right] = \sum \frac{\partial (u_i m\boldsymbol{u})}{\partial x_i} + \frac{\partial m\boldsymbol{u}}{\partial t} \tag{24}$$

$$g\sum_i \left[ \left( \rho \frac{\delta_h}{2} \right)_i \mathrm{proj}_{\perp r}(A_i) \right] = \sum \frac{\partial (u_i \rho V\boldsymbol{u})}{\partial x_i} + \frac{\partial (\rho V\boldsymbol{u})}{\partial t}, \tag{25}$$

which is again inserted into the RHS of the FVE.

$$g\sum_i \left[ \left( \rho \frac{\delta_h}{2} \right)_i \mathrm{proj}_{\perp r}(A_i) \right] = \sum_i \left( q_i \left( \rho n(\boldsymbol{u}_i)\mathbb{1}_{>0}(q_i) \right.\right.$$
$$\left.\left. + \rho \boldsymbol{u}_i \mathbb{1}_{\leq 0}(q_i) \right) \right) + \frac{\partial (\rho V\boldsymbol{u})}{\partial t} \tag{26}$$

The pressure term alone, here on the LHS, constitutes the minimal momentum transport configurations in ocean modeling, as given by the Laplace tidal equations (LTEs) or linearized shallow-water equations (SWEs) (Biewald et al., 2024). With a discrete time derivative and the term for Coriolis acceleration $F_c$, this yields

$$\frac{\rho V \left( \boldsymbol{u}^{+\delta t} - \boldsymbol{u} \right)}{\delta t} = g\sum_i \left[ \left( \rho \frac{\delta_h}{2} \right)_i \mathrm{proj}_{\perp r}(A_i) \right]$$
$$- \sum_i \left( q_i \left( \rho n(\boldsymbol{u}_i)\mathbb{1}_{>0}(q_i) + \rho \boldsymbol{u}_i \mathbb{1}_{\leq 0}(q_i) \right) \right) + F_c. \tag{27}$$

The method is of the first order in space and time to attain high-resolution meshes (Fig. 6) to resolve waves, while remaining efficient in terms of flops: to resolve waves, the cell size should be a log order below the part of the wave spectrum of interest, i.e., maximizing the cell count and minimizing flops per cell.

## 2.3 Coriolis acceleration

Forces, including rotational pseudo-forces, can be substituted into the left-hand side (LHS) of Eq. (20). For the latter, the LHS has to be transformed into the Earth's rotating latitude–longitude reference frame. To observe the acceleration in a rotating reference frame, it can be denoted in terms of the spatial vector relative to the inertial reference frame:

$$\frac{d^2 \boldsymbol{r}}{dt}_i = \left[\frac{\mathrm{d}}{\mathrm{d}t} + \boldsymbol{\Omega} \times\right]\left[\frac{\mathrm{d}\boldsymbol{r}}{\mathrm{d}t} + \boldsymbol{\Omega} \times \boldsymbol{r}\right], \tag{28}$$

where $i$ indicates the inertial reference frame. This yields, for all $n$ components denoted in the momentum vector $m\boldsymbol{u}$,

$$\boldsymbol{F} - m\frac{\mathrm{d}\boldsymbol{\omega}}{\mathrm{d}t} \times \boldsymbol{r} - 2m\left(\boldsymbol{\omega} \times \boldsymbol{v}\right) - m\boldsymbol{\omega} \times \left(\boldsymbol{\omega} \times \boldsymbol{r}\right) = \frac{\partial m\boldsymbol{u}}{\partial t}$$
$$+ \sum\left(u_i \frac{\partial m\boldsymbol{u}}{\partial x_i}\right), \tag{29}$$

with $\boldsymbol{\Omega}$ being Earth's angular velocity. This procedure recovers a term to account for Earth's rotation alone. The effects due to Earth's axial tilt, which is accounted for in the insolation simulation for surface heat exchange, the inclination relative to the solar plane, and the Sun's inclination relative to the galactic plane, are unanimously deemed negligible at the scale of the required transport accuracy and given other uncertainties, such as those due to bathymetric uncertainty. Likewise, Earth's angular velocity is considered a constant, and, hence, its time derivative vanishes:

$$\boldsymbol{F} - 2m\left(\boldsymbol{\omega} \times \boldsymbol{v}\right) - m\boldsymbol{\omega} \times \left(\boldsymbol{\omega} \times \boldsymbol{r}\right) = \frac{\partial m\boldsymbol{u}}{\partial t} + \sum\left(u_i \frac{\partial m\boldsymbol{u}}{\partial x_i}\right). \tag{30}$$

The vertical component of the Coriolis acceleration is deemed negligible (Kundu et al., 2016) and is heavily masked by imperfectly defined vertical turbulent momentum transport. Evaluating the cross products yields, for Earth's zonal and meridional dimensions, a negligible term with quadratic angular velocity, a perpendicular centripetal, and radially inward acceleration; without which

$$\boldsymbol{F} + \begin{pmatrix} 0 & 2m\omega\sin(\phi) & 0 \\ -2m\omega\sin(\phi) & 0 & 0 \\ 0 & 0 & 0 \end{pmatrix}\boldsymbol{u} = \frac{\partial m\boldsymbol{u}}{\partial t}$$
$$+ \sum\left(u_i \frac{\partial m\boldsymbol{u}}{\partial x_i}\right) \tag{31}$$

is obtained for a particular latitude $\phi$ (rad), where Earth's angular velocity $\omega$ (rad s$^{-1}$) is given by $2\pi(24 \times 60^2 \text{ s})^{-1}$. As the Coriolis term does not contain any derivatives, no approximation is required.

## 2.4 Viscous stress, turbulence, LES, and RANS

The consideration of surface forces also accounts for the internal friction of the fluid, the viscosity. Each of the three component velocities at the centroids of the faces of the examined control volume undergoes strain in three spatial dimensions, yielding nine strain rate elements that are commonly presented in tensor form, as shown in Eq. (1). Note that tensor calculus falls here into the confines of matrix calculus. The viscous stress tensor is usually denoted in the form below, including the Nabla operator from the first-order Taylor expansion to attain the differential notation of the force balance at the infinitesimal control volume. For example, for the first component velocity, the dot product yields $\partial\tau_{xx}/\partial x + \partial\tau_{xy}/\partial y + \partial\tau_{xz}/\partial z$ for the first tensor row. The Navier–Stokes PDE is set apart from the Cauchy momentum PDE by being specified for Newtonian fluids where – assuming incompressibility – stress $\tau_{ij}$ is linearly proportional to the sum of the gradient of velocity $i$ in direction $j$ and the gradient of velocity $j$ in direction $i$. The proportionality coefficient $\mu_{ij}$ is termed the viscosity. The entirety of all stresses denoted by the stress tensor can, hence, be substituted by the proportionality of incompressible Newtonian fluids to the sum of the strain rate tensor and its transpose. Note that the absence of volume viscosity is warranted due to the assumption of incompressibility.

$$\nabla \cdot \begin{pmatrix} \tau_{xx} & \tau_{xy} & \tau_{xz} \\ \tau_{yx} & \tau_{yy} & \tau_{yz} \\ \tau_{zx} & \tau_{zy} & \tau_{zz} \end{pmatrix} = \nabla \cdot \left(\begin{pmatrix} \mu_{xx}\frac{\partial u}{\partial x} & \mu_{xy}\frac{\partial u}{\partial y} & \mu_{xz}\frac{\partial u}{\partial z} \\ \mu_{yx}\frac{\partial v}{\partial x} & \mu_{yy}\frac{\partial v}{\partial y} & \mu_{yz}\frac{\partial v}{\partial z} \\ \mu_{zx}\frac{\partial w}{\partial x} & \mu_{zy}\frac{\partial w}{\partial y} & \mu_{zz}\frac{\partial w}{\partial z} \end{pmatrix}\right.$$
$$\left. + \begin{pmatrix} \mu_{xx}\frac{\partial u}{\partial x} & \mu_{yx}\frac{\partial v}{\partial x} & \mu_{zx}\frac{\partial w}{\partial x} \\ \mu_{xy}\frac{\partial u}{\partial y} & \mu_{yy}\frac{\partial v}{\partial y} & \mu_{zy}\frac{\partial w}{\partial y} \\ \mu_{xz}\frac{\partial u}{\partial z} & \mu_{yz}\frac{\partial v}{\partial z} & \mu_{zz}\frac{\partial w}{\partial z} \end{pmatrix}\right) \tag{32}$$

The RHS contains the strain rate tensor and its transpose. In the case of molecular viscosity, that is, momentum transport due to molecular diffusion and interaction, an isotropic and spatially constant coefficient is assumed for all nine $ij$ combinations. As per the latter assumption, the viscosity coefficient can be denoted outside the tensor.

$$\nabla \cdot \mu\left(\begin{pmatrix} \frac{\partial u}{\partial x} & \frac{\partial u}{\partial y} & \frac{\partial u}{\partial z} \\ \frac{\partial v}{\partial x} & \frac{\partial v}{\partial y} & \frac{\partial v}{\partial z} \\ \frac{\partial w}{\partial x} & \frac{\partial w}{\partial y} & \frac{\partial w}{\partial z} \end{pmatrix} + \begin{pmatrix} \frac{\partial u}{\partial x} & \frac{\partial v}{\partial x} & \frac{\partial w}{\partial x} \\ \frac{\partial u}{\partial y} & \frac{\partial v}{\partial y} & \frac{\partial w}{\partial y} \\ \frac{\partial u}{\partial z} & \frac{\partial v}{\partial z} & \frac{\partial w}{\partial z} \end{pmatrix}\right) \tag{33}$$

Inserting term (33) into Eq. (32) and resolving its dot product yields, for the first row (that is, for the viscous stress term of component velocity $u$),

$$\frac{\partial\tau_{xx}}{\partial x} + \frac{\partial\tau_{xy}}{\partial y} + \frac{\partial\tau_{xz}}{\partial z} = \mu\left(\frac{\partial\left(\frac{\partial u}{\partial x} + \frac{\partial u}{\partial x}\right)}{\partial x} + \frac{\partial\left(\frac{\partial u}{\partial y} + \frac{\partial v}{\partial x}\right)}{\partial y}\right.$$
$$\left. + \frac{\partial\left(\frac{\partial u}{\partial z} + \frac{\partial w}{\partial x}\right)}{\partial z}\right). \tag{34}$$

In Eulerian fluid dynamics, that is, continuum mechanics, the assumption of continuous variables flows directly from

the very concept under consideration and is assumed for the quantities' derivatives as well. Therefore, Clairaut's theorem can be applied, rendering the order of partial differentiation immaterial. This assumption breaks down at quantity jumps. Yet at infinite gradients, Eulerian diffusive models break down anyway. Fortunately, such conditions are, at the scale considered, not present in the coastal ocean systems described here. Therefore, the partial derivatives on the LHS can be sorted as follows:

$$\frac{\partial \tau_{xx}}{\partial x} + \frac{\partial \tau_{xy}}{\partial y} + \frac{\partial \tau_{xz}}{\partial z} \mu \left( \frac{\partial^2 u}{\partial x^2} + \frac{\partial^2 u}{\partial y^2} + \frac{\partial^2 u}{\partial z^2} \right)$$
$$+ \mu \frac{\partial \overbrace{\left( \frac{\partial u}{\partial x} + \frac{\partial v}{\partial y} + \frac{\partial w}{\partial z} \right)}^{=0}}{\partial x}. \tag{35}$$

Here the insertion of the continuity PDE $\nabla \boldsymbol{u} = 0$ eliminates three of the partial derivatives, yielding, for the entire system of PDEs,

$$\frac{\partial (\rho \, \boldsymbol{u} \, V)}{\partial t} + \sum \frac{\partial (u_i \rho_i \, \boldsymbol{u} \, V)}{\partial x} = \boldsymbol{F}$$
$$+ \mu \left( \frac{\partial^2 \boldsymbol{u}}{\partial x^2} + \frac{\partial^2 \boldsymbol{u}}{\partial y^2} + \frac{\partial^2 \boldsymbol{u}}{\partial z^2} \right). \tag{36}$$

Molecular viscosity is isotropic and largely constant and can, therefore, due to being constant, be placed outside the derivative. Yet eddy viscosity is not at all constant. In the case of eddy viscosity, the Smagorinsky model assumes horizontally isotropic viscosity. Furthermore,

$$\frac{\partial w}{\partial x} \ll \frac{\partial u}{\partial x}, \quad \frac{\partial w}{\partial x} \ll \frac{\partial u}{\partial y}, \quad \frac{\partial w}{\partial x} \ll \frac{\partial v}{\partial x} \tag{37}$$

and analogously for $\partial w / \partial y$, yielding

$$\nabla \cdot \left( \begin{pmatrix} k\frac{\partial u}{\partial x} & k\frac{\partial u}{\partial y} & k_z\frac{\partial u}{\partial z} \\ k\frac{\partial v}{\partial x} & k\frac{\partial v}{\partial y} & k_z\frac{\partial v}{\partial z} \\ k\frac{\partial w}{\partial x} & k\frac{\partial w}{\partial y} & k_z\frac{\partial w}{\partial z} \end{pmatrix} + \begin{pmatrix} k\frac{\partial u}{\partial x} & k\frac{\partial v}{\partial x} & 0\frac{\partial w}{\partial x} \\ k\frac{\partial u}{\partial y} & k\frac{\partial v}{\partial y} & 0\frac{\partial w}{\partial y} \\ k_z\frac{\partial u}{\partial z} & k_z\frac{\partial v}{\partial z} & k_z\frac{\partial w}{\partial z} \end{pmatrix} \right), \tag{38}$$

with horizontally isotropic eddy viscosity $k$ and vertical eddy viscosity $k_z$. Therefore, Clairaut's theorem cannot be applied, except to the third row of the transpose. That is, the transpose of the strain rate tensor remains relevant. The partial derivatives are, thus, collected differently, yielding, for the horizontal component velocities,

$$\frac{\partial (\rho \, u \, V)}{\partial t} + \sum \frac{\partial (u_i \rho_i \, u \, V)}{\partial x} = \boldsymbol{F} + 2\frac{\partial (k \partial u / \partial x)}{\partial x}$$
$$+ \frac{\partial (k (\partial u / \partial y + \partial v / \partial x))}{\partial y} + \frac{\partial (k_z \partial u / \partial z)}{\partial z} \tag{39}$$

$$\frac{\partial (\rho \, v \, V)}{\partial t} + \sum \frac{\partial (u_i \rho_i \, v \, V)}{\partial x} = \boldsymbol{F} + 2\frac{\partial (k \partial v / \partial y)}{\partial y}$$
$$+ \frac{\partial (k (\partial v / \partial x + \partial u / \partial y))}{\partial x} + \frac{\partial (k_z \partial v / \partial z)}{\partial z}. \tag{40}$$

The dot product of the transpose of the strain rate tensor for the third row, the vertical velocity component $w$, is

$$\nabla \cdot \left( k\frac{\partial w}{\partial x} + k\frac{\partial w}{\partial y} + k_z\frac{\partial w}{\partial z} \right) + \frac{\partial \left( k_z \frac{\partial u}{\partial z} \right)}{\partial x}$$
$$+ \frac{\partial \left( k_z \frac{\partial v}{\partial z} \right)}{\partial y} + \frac{\partial \left( k_z \frac{\partial w}{\partial z} \right)}{\partial z}. \tag{41}$$

Applying the product rule and sorting terms produces

$$\nabla \cdot \left( k\frac{\partial w}{\partial x} + k\frac{\partial w}{\partial y} + k_z\frac{\partial w}{\partial z} \right) + \frac{\partial k_z}{\partial x}\frac{\partial u}{\partial z} + \frac{\partial k_z}{\partial y}\frac{\partial v}{\partial z}$$
$$+ \frac{\partial k_z}{\partial z}\frac{\partial w}{\partial z} + k_z \left( \frac{\partial \frac{\partial u}{\partial z}}{\partial x} + \frac{\partial \frac{\partial v}{\partial z}}{\partial y} + \frac{\partial \frac{\partial w}{\partial z}}{\partial z} \right). \tag{42}$$

For the last three terms, Clairaut's theorem can again be applied, changing the order of partial differentiation. Furthermore, some terms are approximated using finite differences.

$$\nabla \cdot \left( k\frac{\partial w}{\partial x} + k\frac{\partial w}{\partial y} + k_z\frac{\partial w}{\partial z} \right) + \frac{\delta k_z}{\delta x}\frac{\delta u}{\delta z} + \frac{\delta k_z}{\delta y}\frac{\delta v}{\delta z}$$
$$+ \frac{\delta k_z}{\delta z}\frac{\delta w}{\delta z} + k_z \frac{\partial \overbrace{\left( \frac{\partial u}{\partial x} + \frac{\partial v}{\partial y} + \frac{\partial w}{\partial z} \right)}^{=0}}{\partial z}, \tag{43}$$

again yielding the opportunity to exploit the continuity PDE to eliminate terms. Also, the finite difference approximation yields the opportunity to rearrange divisors:

$$\nabla \cdot \left( k\frac{\partial w}{\partial x} + k\frac{\partial w}{\partial y} + k_z\frac{\partial w}{\partial z} \right) + \frac{\delta k_z}{\delta z}\overbrace{\left( \frac{\delta u}{\delta x} + \frac{\delta v}{\delta y} + \frac{\delta w}{\delta z} \right)}^{\approx 0}, \tag{44}$$

recovering the finite difference approximation of the continuity PDE, which is approximately zero, and, thus, eliminating further terms. Inserted into the PDE for the vertical component velocity, this yields

$$\frac{\partial (\rho \, w \, V)}{\partial t} + \sum \frac{\partial (u_i \rho_i \, w \, V)}{\partial x} = \boldsymbol{F} + \frac{\partial (k \partial w / \partial x)}{\partial x}$$
$$+ \frac{\partial (k \partial w / \partial y)}{\partial y} + \frac{\partial (k_z \partial w / \partial z)}{\partial z}. \tag{45}$$

Tables 1 and 2 list the selectable approximations for the eddy diffusive terms. Transport unresolved by the mesh is at the smallest scale of molecular diffusion that is also present during laminar flow. Negligible molecular diffusion and also unresolved eddies are usually modeled as random, that is, diffusive phenomena, and are referred to by the fictitious quantity eddy diffusion. Direct numerical simulation (DNS) is computationally prohibitively expensive on the geophysical scale without a quantum computing resource (Itani, 2021; Bharadwaj and Sreenivasan, 2022). Henceforth, unresolved turbulence is treated

using two fictitious models and quantities: (1) Reynolds-averaged Navier–Stokes (RANS) simulations and (2) large-eddy simulations (LES). In the first approach, transient fluctuations are split off the modeled velocity, whereas in the second, transport is filtered by deducting (Smagorinsky, 1963) spatially unresolved transport. That is, the former smooths over time, the latter smooths over space, and both can be used in the same model for the vertical and horizontal, respectively (Yu et al., 2017, 2018).

The RANS approach is more receptive to buoyancy, which is important for vertical turbulence, whereas LES, developed by Smagorinsky, is computationally more effective for isotropic flows, which are a valid description of the horizontal. Therefore, RANS models, in particular $k$–$\epsilon$ models, are commonly used for the vertical, and an LES–Smagorinsky model is used for the horizontal. The same approach has been selected for this model and documented in subsequent sections.

The transport PDEs derived above, including a diffusive term for turbulence in conjunction with a density quantifying PDE, are termed the primitive equations, where primitive corresponds to its basic governing utility and not to a lack of sophistication. Or, if inertial considerations are restricted to two dimensions and the vertical is approximated by mere continuity, then the set of PDEs that is yielded is called shallow-water equations.

## 2.5 Smagorinsky turbulence

Transport of vector and scalar quantities is split into a resolved and an unresolved component. This step is commonly referred to as filtering, with a filtered velocity $\overline{u}$ and an unresolved residual velocity. Underlying the Smagorinsky model is the assumption that Reynolds stress can be modeled by the rate of the strain tensor, introducing another fictitious viscosity that is itself modeled by the rate of the strain tensor. The Smagorinsky model has an inherent assumption of an even weighting of the fluctuating velocity. The averaging in the filter function can also introduce a heavier weight close to the centroids, given that the velocities are stored at the centroid.

The Smagorinsky model corresponds only to a uniform isotropic filter, also termed a box filter. An alternative is a Gaussian, bell-shaped filter function that samples the fluctuation velocity predominantly at the centroids. The eddy viscosity is dependent on the filter size, here the Voronoi cell size. Molecular viscosity is usually ignored, given that it is logarithmic orders of magnitude smaller than eddy viscosity. The Smagorinsky model is simpler than the RANS model, as it assumes isotropic turbulence, which is warranted in the horizontal plane but not in the vertical plane.

In the Smagorinsky model, the strain rate tensor of the resolved flow represents the local deformation of the flow. The model is obtained from Kolmogorov modeling (Kolmogorov, 1941) of the turbulent energy dissipation from the average

of the fluctuating velocity, where the proportionality of the former to the Reynolds stress is inferred from dimensional analysis. Utilization of a diffusive term under the assumption of random and, thus, gradient-driven transport is readily obtained from the fluid's Lagrangian description, given that random transport holds in the thermodynamic Lagrangian domain at every scale of unresolved geometry.

$$k_{\mathrm{H}} = \frac{C_{\mathrm{S}} P}{2} \sqrt{\left(\frac{\partial u}{\partial x}\right)^2 + \left(\frac{\partial v}{\partial y}\right)^2 + 2^{-1}\left(\frac{\partial v}{\partial x} + \frac{\partial u}{\partial y}\right)^2}, \quad (46)$$

with the Smagorinsky coefficient $C_{\mathrm{S}}$. The eddy diffusivity constant is usually (Yang and Khangaonkar, 2008; Chen et al., 2011) treated as isotropic horizontally but is computed separately for the vertical. The eddy diffusivity is modeled with the eddy viscosity $\mu$ for momentum transport, which can be set as prognostic or diagnostic for the horizontal and the vertical. That is, in the latter case, a value obtained from a parameter identification is applied. Otherwise, in the prognostic case, the horizontal eddy viscosity is computed using the Smagorinsky model.

Numerical diffusion is to be expected as well, that is, artificial diffusion inherent to continuum descriptions of physics. Numerical diffusion does not pose a detriment to the stability but does to the accuracy of a simulation, by inflating eddy diffusion. Nevertheless, LES and RANS models permit nominal compliance with the Navier–Stokes PDE, comprehensiveness in approximating all terms, or possible stability benefits that a diffusive term conveys to some algorithms.

The evaluation of the Cartesian gradient at the centroids is not readily available in unstructured meshes and requires dedicated computation (Skamarock et al., 2012). Given that the magnitudes of other uncertainties are of higher logarithmic orders in the LES and RANS models, it appears questionable to expend flops to refine the computation of the derivatives in Eq. (46) to compute estimates for this fictitious quantity. Nevertheless, to not have to quantify the uncertainty due to these interpolations, the derivatives of the velocity components at the centroid are here rigorously computed by applying the total derivative to each triangle that spans a centroid and two adjacent neighbors. Most of these computations can be calculated ahead of the simulation and stored in auxiliary arrays, as provided in Table 2.

The filtered transport in the eddy diffusive terms requires a proportionality coefficient to link the strain rate to an eddy diffusive force. In coastal ocean momentum transport, it is commonly assumed that this coefficient grows with the LES spatial filter, represented by the Voronoi polygon size $P$, and with the strain rate terms in the root of Eq. (46) above. Algorithms to compute gradients are listed in Tables 1 and 2. All geometric meta-quantities are calculated in advance ahead of the simulation and are stored to minimize the computational load.

## 2.6 Hydrodynamic boundary conditions

The no-flux boundary is simply attained by setting all fluxes through the boundary face to zero. For scalar quantities, that is, quantities without a hydrostatic pressure term such as temperature, salinity, sediment, and tracers, the free-slip boundary condition keeps quantities from growing out of bounds. The free-slip boundary condition keeps the velocity vector in the finite volume parallel to the boundary. That is, no component is perpendicular to a solid boundary face. At the open boundary conditions that connect the model to the open sea, the surface elevation is set to the surveyed tidal meter time series or to tidal data from any other source.

Rigor in depicting bottom stress is limited by the usually unknown roughness distribution of the vast seafloor. Some assumptions have been made here: without a velocity to shed from or fluid to interact with, no friction force can act, which renders the assumption of bottom stress being dependent on velocity and density plausible. If an advected fluid particle is exerted upon by isotropically oriented roughness elements, then the number of exertions is proportional to the advection velocity's magnitude and Lagrangian particle density. Furthermore, the net force exerted can, as with any force, be denoted by its components, that is, in terms of velocity components, as momentum is removed from the same. Therefore, the bottom-stress proportionality is obtained and is rendered as an equation (Feddersen et al., 2003; Faria et al., 1998) by introducing a proportionality coefficient:

$$\boldsymbol{\tau} \propto \rho \langle \|\boldsymbol{u}\| \boldsymbol{u} \rangle$$
$$\boldsymbol{\tau} = \rho c_{\mathrm{d}} \langle \|\boldsymbol{u}\| \boldsymbol{u} \rangle, \tag{47}$$

with the coefficient $c_{\mathrm{d}}$ termed the drag coefficient. As evidenced by Eq. (47) for a particular stress $\boldsymbol{\tau}$, the drag coefficient $c_{\mathrm{d}}$ depends on the height above the bottom, given the velocity's dependence on this height. For example, if a slim bottom layer is modeled utilizing a slow segment within the vertical velocity profile, then the drag coefficient grows.

An estimate for $c_{\mathrm{d}}$ is usually obtained (Grant and Madsen, 1979) by applying the Prandtl–Kármán logarithmic velocity profile to oceanic applications, that is,

$$\|\boldsymbol{u}\| = \frac{u_*}{\kappa} \ln \left( \frac{z}{z_0} \right), \tag{48}$$

where the friction velocity is defined by

$$\|\boldsymbol{\tau}\| = \rho u_*^2, \tag{49}$$

with the bottom stress $\boldsymbol{\tau}$, the friction velocity $u_*$, the Kármán constant $\kappa = 0.4$, the roughness length $z_0$, and the height above the seafloor $z$. Bottom drag has been specified by conducting parameter identifications for the friction velocity and drag coefficient (Faria et al., 1998) or by identifying a suitable roughness length (Isobe and Beardsley, 2006; Weisberg and Zheng, 2006; Yang and Khangaonkar, 2008). Yet, while the drag coefficient depends on the bottom-layer height, the friction velocity depends on the local velocity. Both are, thus, not independent of transient quantities. Therefore, for the purpose of modeling various depths and flow regimes, only the roughness length is sufficiently fundamental to hold for the entire tidal cycle. Resolving Eq. (48) for $u_*$ and specifying the RHS for two heights eliminate the friction velocity:

$$\frac{\kappa \|\boldsymbol{u}_1\|}{\ln \left( \frac{z_1}{z_0} \right)} = \frac{\kappa \|\boldsymbol{u}_2\|}{\ln \left( \frac{z_2}{z_0} \right)}, \tag{50}$$

and, after rearrangement, this resolves to the ubiquitous form for roughness length identification in atmospheric and oceanic applications alike:

$$\ln z_0 = \frac{\|\boldsymbol{u}_1\| \ln z_2 - \|\boldsymbol{u}_2\| \ln z_1}{\|\boldsymbol{u}_1\| - \|\boldsymbol{u}_2\|}. \tag{51}$$

The drag coefficient $c_{\mathrm{d}}$ in the boundary condition is then obtained by taking the magnitude of the LHS and RHS vectors in Eq. (47) while substituting the RHS from Eq. (49) for the LHS.

$$\rho u_*^2 = \|\rho c_{\mathrm{d}} \langle \|\boldsymbol{u}\| \boldsymbol{u} \rangle \| \tag{52}$$

Furthermore, Eq. (48), resolved for the friction velocity, substitutes for the same term in Eq. (52).

$$\left( \frac{\kappa \|\boldsymbol{u}\|}{\ln z - \ln z_0} \right)^2 = c_{\mathrm{d}} \|\boldsymbol{u}\| \|\boldsymbol{u}\|, \tag{53}$$

thus yielding the common form (Chen et al., 2011) for the calculation of the drag coefficient distribution from the roughness length,

$$c_{\mathrm{d}}(x, y, t) = \left( \frac{\kappa}{\ln z(x, y, t) - \ln z_0} \right)^2. \tag{54}$$

In Eq. (54), it is explicit that $c_{\mathrm{d}}(x, y, t)$ is not a constant but a horizontal distribution if the layer thickness $z(x, y, t)$ is not constant. Seabed change or tidal transience, depending on the layer configuration, can result in a time dependency.

## 2.7 Sediment transport, waves, and entrainment

Sediment is transported like any other constituent, with the addition of sediment settling and bottom-sediment entrainment. A mass balance for the sediment type $C$ returns

$$\frac{\partial C}{\partial t} + \frac{\partial (uC)}{\partial x} + \frac{\partial (vC)}{\partial y} + \frac{\partial ((w - w_{\mathrm{s}})C)}{\partial z}$$
$$= \frac{\partial}{\partial x} \left( D_x \frac{\partial C}{\partial x} \right) + \frac{\partial}{\partial y} \left( D_y \frac{\partial C}{\partial y} \right) + \frac{\partial}{\partial z} \left( D_z \frac{\partial C}{\partial z} \right) - E, \tag{55}$$

with the entrainment term $E$. The consideration of eddy diffusion in the sections above has shown that the horizontal

eddy diffusive coefficients are horizontally isotropic. Furthermore, the insertion of continuity renders the velocities outside of the derivatives.

$$\frac{\partial C}{\partial t} + u\frac{\partial C}{\partial x} + v\frac{\partial C}{\partial y} + (w - w_s)\frac{\partial C}{\partial z} = \frac{\partial}{\partial x}\left(A_h\frac{\partial C}{\partial x}\right)$$
$$+ \frac{\partial}{\partial y}\left(A_h\frac{\partial C}{\partial y}\right) + \frac{\partial}{\partial z}\left(D_z\frac{\partial C}{\partial z}\right) - E \tag{56}$$

The settling velocity $w_s$ for fine sediment, set in the input file, can be calculated with Stoke's law:

$$w_s = \frac{d^2 g(\rho_s - \rho)}{18\mu}, \tag{57}$$

where $d$ is the diameter of the sediment particle (m), $g$ is the acceleration due to gravity ($\mathrm{m\,s^{-2}}$), $\rho_s$ is the density of the sediment particle ($\mathrm{kg\,m^{-3}}$), $\rho$ is the density of the fluid ($\mathrm{kg\,m^{-3}}$), and $\mu$ is the dynamic viscosity of the fluid ($\mathrm{kg\,m^{-1}\,s^{-1}}$). The influence of orbital wave motion on near-bottom tidal currents depends on the length of the agitating waves. If conditions are calm, with wind waves exhibiting wavelengths much smaller than twice the water depth, then the perturbation of near-bottom tidal currents due to orbital wave motion remains small or insignificant. If conditions are sufficiently agitated, such that wavelengths reach the order of magnitude of the water depth in size, then wave orbital motion drives bottom currents considerably. Additionally, high-energy waves yield disproportionate increases in erosive fluxes, contributing to shoreline development considerably. The Rouse number that indicates whether sediment entrainment or deposition occurs is given by

$$Ro = \frac{w_s}{\kappa u_*}. \tag{58}$$

For sheltered conditions, the survey's wave meter recorded wavelengths not exceeding 0.5 m. Therefore, erosive fluxes have been simulated for three conditions, with a dedicated high-resolution mesh for wave-resolving simulations, as detailed in Sect. 3.4: tidal currents during sheltered conditions and during two high-energy-wave scenarios are shown. Waves are approximated as second-order Stokes waves as per the Le Méhauté diagram.

## 3 Application

Documentation of how to build the model step by step is given in the subsections on meshing (Sect. 3.1) and on case-dependent horizontal and vertical boundary conditions (Sect. 3.2) below. That is, setting tidal boundary conditions to drive the model and setting the bottom-friction parameter to attenuate it are shown. The water body modeled is shown in Fig. 1. A highly resolved beach model for wave simulations is nested within a model of the entirety of the bay. Five survey locations in Doha Bay provided boundary forcing (two) and triple validation, in addition to examining the same locations for two different seasons.

### 3.1 Meshing

To obtain the horizontal geometry of the sea surface, a satellite image, as exhibited in Fig. 1, can be downloaded from Google Earth; however, any other image, map, or CAD drawing can be used. The coastline and boundaries are then marked with a 24-bit RGB code identifier in a .bmp file in any .bmp editing tool. CE2All land pixels are then automatically flood-filled after setting all other pixels to 0 using the logical array and flood-fill function `excise('image.bmp',RGB)` with a particular 24-bit `RGB` color, that is, with maximal component values of `[255 255 255]`.

Maps and CAD designs of future developments can be superimposed, using the script `overlay`. The mesh is created directly from the .bmp using the mesh generator `meshing22a('image.bmp')`. The latter automatically provides a higher resolution at the boundary between land and sea by first distributing Voronoi polygon seeds by sweeping along the shore with increasing distance, followed by three iterations of mesh relaxation. The relaxation algorithm redistributes the polygon centroids as per Lloyd's algorithm but is based on a discrete tessellation. The mesh depicted in Fig. 2 is georeferenced by marking two reference coordinates, $(x_a, y_a)$ and $(x_b, y_b)$, within the 24-bit .bmp file fed to the meshing generator with a particular 24-bit `RGB color`. The function `[x1,y1,xd,yd] = coord23('.bmp', RGB,xa,ya,xb,yb)` then returns the coordinates of the bottom-left corner $(x1, y1)$ of the image and the first reference point $(xa, ya)$ in pixel coordinates $(xd, yd)$. These can be used to scale and translate the mesh to georeference it for small sites.

The georeferencing is embedded within the script for the bathymetry interpolation onto the mesh, `bath22a`, by book-keeping a depth value in the vector `hb` for each cell. The interpolation of `bath22a` attributes surveyed and remotely sensed depths according to their area share of Voronoi polygons. The composite of the surveyed and remotely sensed bathymetry is shown in Fig. 2.

A string of finite volume cells that bound mangrove nooks, ongoing developments, coverage by marine vessels, or ill-resolved harbor bathymetry can be identified in an index plot with `plot0(u,v,1:length(u),'mesh.mat','0', '-','txt')`; null vectors as dummy velocities; and the visualized quantity, here the indices, denoted (`'txt'`) in each delineated (`'-'`) cell. The meshing code is not discussed here because the consistency of the output Voronoi diagram can be ascertained visually.

The boundary of the area to be amended is then denoted in a list of cells (`list_const`) with a constant null concentration (`c_const = 0`) in the simulation's case file (`c(list_const) = c_const`). Any random cell inside a listed area of concern (`list_c0`) can conveniently return all the area's cells as non-zero via advective propagation. That is, the tidal advection simulation is

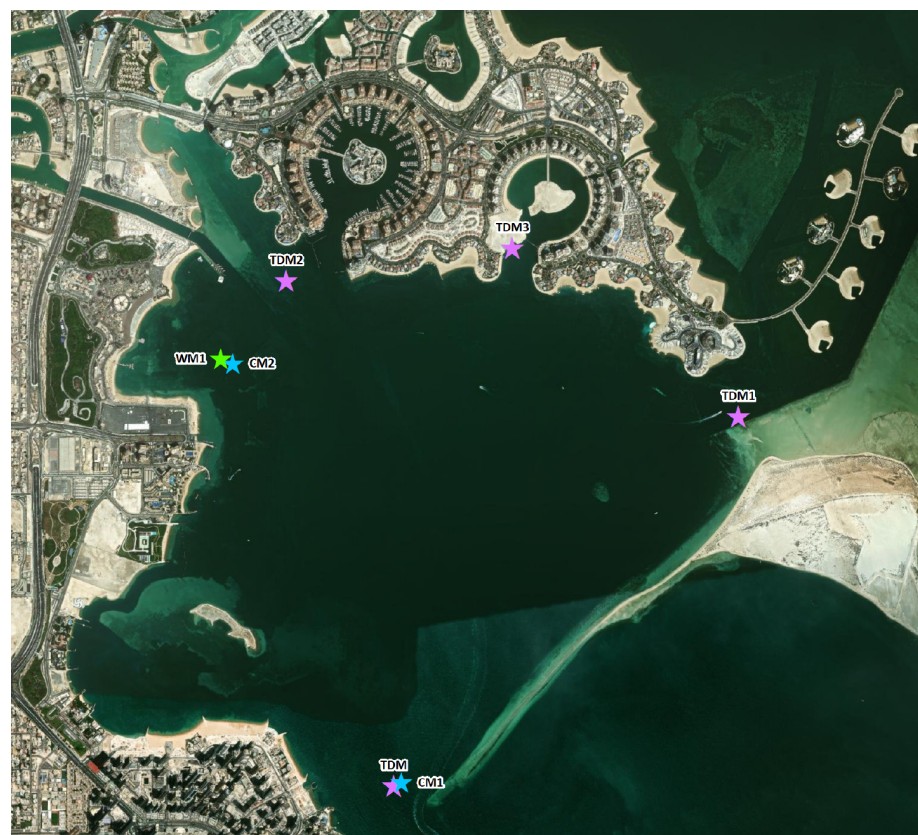

**Figure 1.** Doha Bay, © Google Earth 2023. The stars indicate the locations of the tidal and current meters.

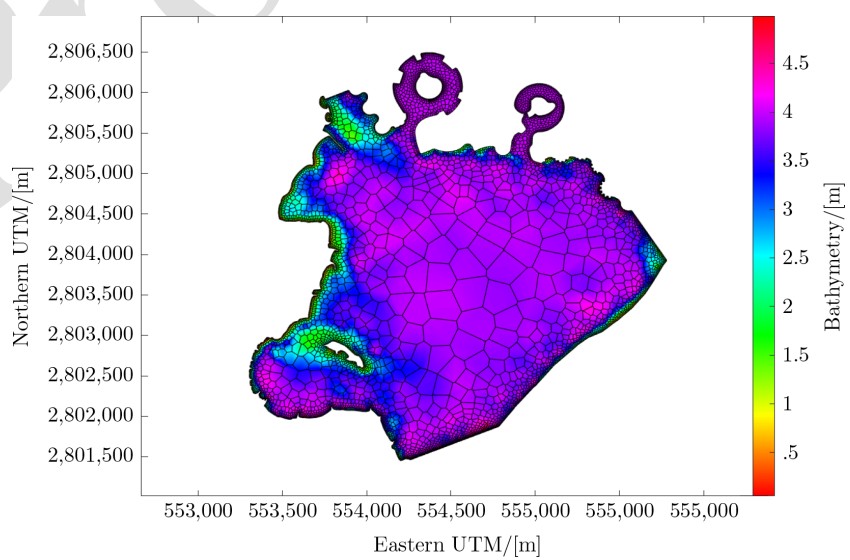

**Figure 2.** Bathymetry with corrected marina design depth. The wave-resolving mesh is shown in Fig. 6.

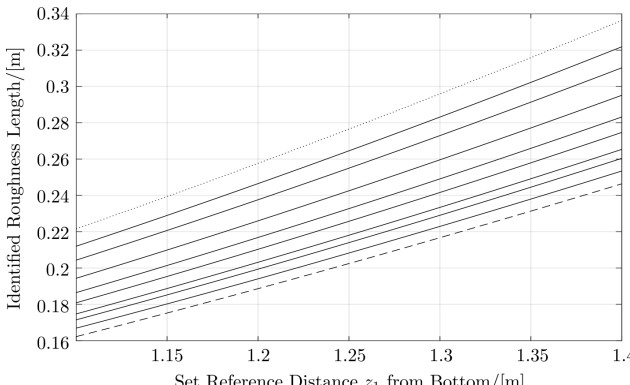

**Figure 3.** The identified roughness length $z_0$ vs. varied uncertain reference height $z_1 = 1.25 \pm 0.15$ m and the altered minimal velocity difference $\delta u$ from 1 % (dashed line) to 10 % (dotted line), with ascending 1 % increments in between (solid lines).

merely exploited to automatically mark the area delineated by `list_const`. Identified cells, that is, a concerned area `area_i = (c>0)`, may be bookkept `save name.mat area_i` and set (`hb(area_i) = `) to the desired depth, in this case, 4 m for the harbor and 1 m for the mangrove forest.

### 3.2  Boundary conditions

Tidal, current, open, and/or other horizontal boundaries are specified by plotting cell indices with the plotting function accordingly configured, using a zero-order interpolation `plot23(u,v,1:length(u),'mesh.mat','0', '-','txt')`. The mesh generator numbers cell indices sequentially along the boundary. Therefore, quantities at the boundary can be set to boundary conditions by referring to a particular boundary section with `index_1:index_2`. For example, in the case of a tidal boundary condition, `nodes(1:length(istart:iend)) = istart:iend`.

The bed roughness length was identified, as illustrated in Fig. 3, with Eq. (51) used to specify the bottom-boundary condition. The time series of vertical profiles from current meters came with some uncertainties: occasionally the surface velocity is slower than in lower layers, which can occur due to transient dynamics such as wind forcing. The vertical profile spacing from the sea surface, with a top layer of 1.2 m and other layers of 0.5 m, exhibited cumulative layer thicknesses that occasionally did not match the total measured depth. Therefore, considerable uncertainty in the layer thickness had to be assumed, and a sensitivity study was conducted for the same.

Equation (51) can be ill-suited to such uncertainties, which will not always average out: if, for example, there is only a minuscule difference between the two velocities in the denominator, then the roughness length is overestimated by

logarithmic orders of magnitude. Likewise, the equation is sensitive to an error in height $z_1$. Consequentially, the differences between layers that are two layers apart, that is, layer 3 and 5, instead of adjacent 0.5 m thin layers have been considered. The bottom layer, in principle, would have been more indicative but has been disregarded to obtain a lower relative error for the reference height. Additionally, measurements were excluded that did not exhibit a significant or positive vertical velocity difference $\delta u = u_1 - u_2$.

In order to obtain a well-behaved response despite the uncertainties in reference height and the filtration of small $\delta u$, both parameters have been varied, and the parameter identification was conducted with 2000 measurements. The roughness length distribution is shown in Fig. 3. Compared to the considerable fluctuations in surface friction in particular, Fig. 3 retains the good behavior and returns a roughness length on the order of 0.2 m, regardless of the minimal $\delta$ in velocity and the assumed reference height. That is, regardless of the two parameters varied, a roughness length on the order of 0.2 m is obtained.

### 3.3  Validation

Surface elevation predictions have been stored during the simulation for finite volumes that correspond to the tidal and current meter locations within the computational domain. The measured and simulated time series were vertically referenced to the observed mean sea level with `TM = TM - mean(TM)` and were correlated. For the measured time series, the start time is specified using `t1 = datetime(2022,9,14,12,0,0)`, and the end time is translocated by `hours(.5)*(length(TM)-1)` with respect to the start time. The resultant time vector is built as per the data point frequency `tTM=t1:hours(.5):t2`, reflective of the half-hourly sampling. The trivial `hold on` command allows the superposition of the plots for the measured and simulated surface elevation with `plot(tTM,TM)` and so forth.

Surveyed and simulated surface elevations, as well as the error, mean error, and root-mean-square error (RMSE), are depicted in Figs. 4 and 5. Table 3, furthermore, contains percent errors besides absolute errors: percent RMSE, $R^2$, and Pearson's $R$ (Reese et al., 2024; Barghorn et al., 2024), to quantify the quality of the correlation. Two survey locations served to specify the boundary forcing, and three survey locations served to validate the accuracy of the simulation. Given that two seasons have been examined, four time series were available for boundary forcing and six to validate the simulation.

### 3.4  Waves, sediment entrainment, and settling

Wave motion has been resolved with a high-resolution mesh for wave propagation, with an even resolution throughout the entire domain. High-energy waves, according to the National

**Table 3.** Validation of the simulation with measured surface elevation.

| Quantity | CM2 August | TM2 August | TM3 August | TM2 April | TM3 April |
|---|---|---|---|---|---|
| % error $\epsilon$ | 1.7 | 4.8 | 2.4 | 5.6 | 3.8 |
| Abs. $\epsilon$ [cm] | 3.3 | 9.4 | 4.6 | 8.9 | 6.0 |
| RMSE [cm] | 4.4 | 12 | 5.9 | 10 | 7.6 |
| % NRMSE | 2.3 | 6.9 | 3.1 | 6.5 | 4.8 |
| $R^2$ | 0.99 | 0.94 | 0.98 | 0.94 | 0.97 |
| Pearson's $R$ | 1.0* | 0.97 | 0.99 | 0.97 | 0.99 |

* 0.9957 rounded to the third digit.

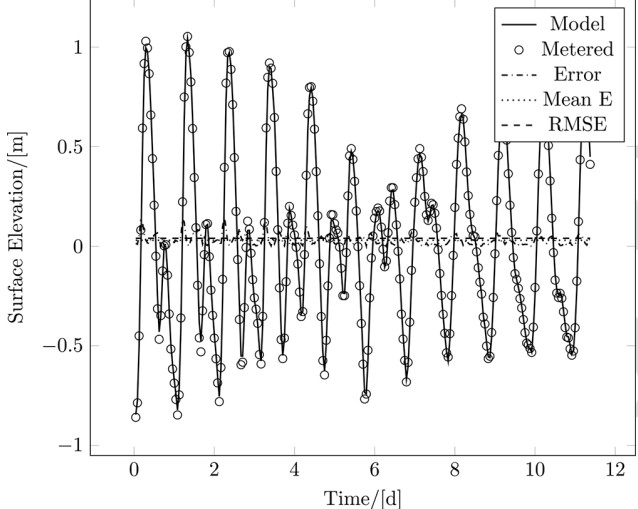

**Figure 4.** Correlation between simulated and surveyed surface elevation at the location of current meter no. 2, which also recorded depth, that is, surface elevation, in August 2023.

Oceanic and Atmospheric Administration (NOAA) CFSR model, have been compiled (Lawen, 2024b) for 25.50° N, 52.17° E, listed in Table 4, and wave transmission has been modeled within Doha Bay, as cited by Lawen (2024b). For 5 the shortest and longest wave periods, 6.40 and 7.12 s, a transmission to significant wave heights on the order of 0.1 and 0.4 m, respectively, was found (Lawen, 2024b). The same parameters were simulated with the wave-resolving model in order to resolve wave transmissions for local 10 beaches. The periods of 6.4 and 7.12 s correspond to wavelengths $L = gT^2/(2\pi)$ of 64 and 79 m, which were resolved using the 6 m fine, high-resolution mesh without wetting and drying (Memmola et al., 2020).

Fine structures (Fig. 7) in tidal currents have also been 15 resolved using the high-resolution mesh. Both horizontal

**Table 4.** High-energy wave conditions east of Safliya Island, NOAA CFSR.

| Return period [yr] | $H_s$ [m] | $T_p$ [s] | Direction [°] |
|---|---|---|---|
| 100 | 1.92 | 6.40 | 12.5 |
| 100 | 2.34 | 7.12 | 57.5 |
| 100 | 2.67 | 6.99 | 102.5 |
| 100 | 2.05 | 5.72 | 147.5 |

geometry and bathymetry determine the transformation of incoming waves. Acceleration due to continuity at narrow or shallow sections yields acceleration in both the depth-averaged and the friction velocity, the latter driving the entrainment of sediment and, hence, erosion. 20 Second-order Stokes waves enter the high-resolution domain, which is depicted for the friction velocity in Fig. 8. The boundary of the high-resolution domain is aligned with the wave direction (Lawen, 2024b), with the waves superimposed on the tidal boundary condition. High friction veloc- 25 ity entails erosion. The dynamic friction velocity distribution can, thus, reveal spots that are vulnerable to morphological changes.

The dynamic Rouse number distribution visualizes where and when sediment settling and erosion predominate, as both 30 are time-dependent (Patzke et al., 2022). Values below and above 1 correspond to erosion and settling, respectively. Island developments perturb the natural coastal ocean equilibrium of sediment transport. The highly resolved simulation (Fig. 9) brings into focus the fine pattern and structures, en- 35 hancing the reliability of the former based on mitigating measures.

The Rouse number distribution was simulated, resolving wavelengths on the order of 60 m on a 6 m Voronoi mesh, avoiding wave fronts on acute finite volume polygon angles. 40 The tile pattern exhibited in Fig. 6 stems from the 30 m resolution of the open-source Landsat images utilized in remote sensing. The mesh thus has a higher resolution than the bathymetric model. Nevertheless, commercially available satellite images can facilitate a remote sensing resolution that 45 matches the resolution of the mesh.

A high-resolution mesh was produced, and the wave boundary was aligned with the wave direction. Currents were simulated for sheltered and high-energy conditions, and the Rouse number distribution is shown for the latter. If the 50 wave's orbital motion does not reach the seafloor, then perturbations of bottom-layer currents are small. But waves with wavelengths in excess of the water depth do exert an influence on bottom currents, with the latter governing shear and, thus, erosion. Such medium- and long-wavelength waves can 55 result from tidal forcing, displayed in Fig. 7 as seiches and longwave agitation.

The friction velocity distribution (Fig. 8) and the ratio between settling and erosion fluxes (Fig. 9) match previously

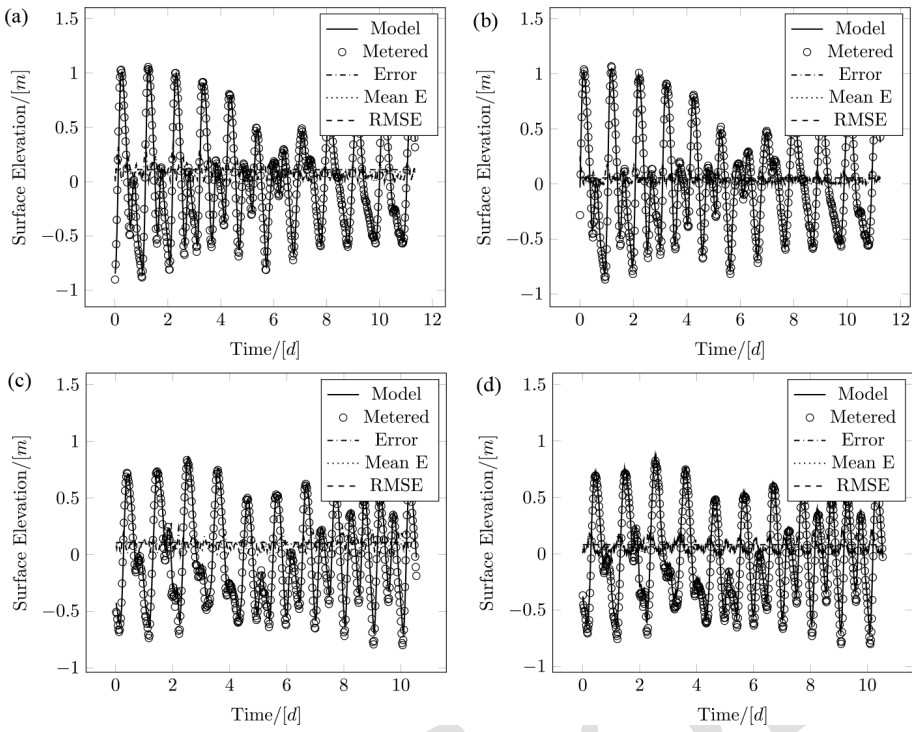

**Figure 5.** Correlation between simulated and surveyed surface elevation at the location of tidal meters no. 2 **(a, c)** and no. 3 **(b, d)** during August **(a, b)** and April **(c, d)** of 2023.

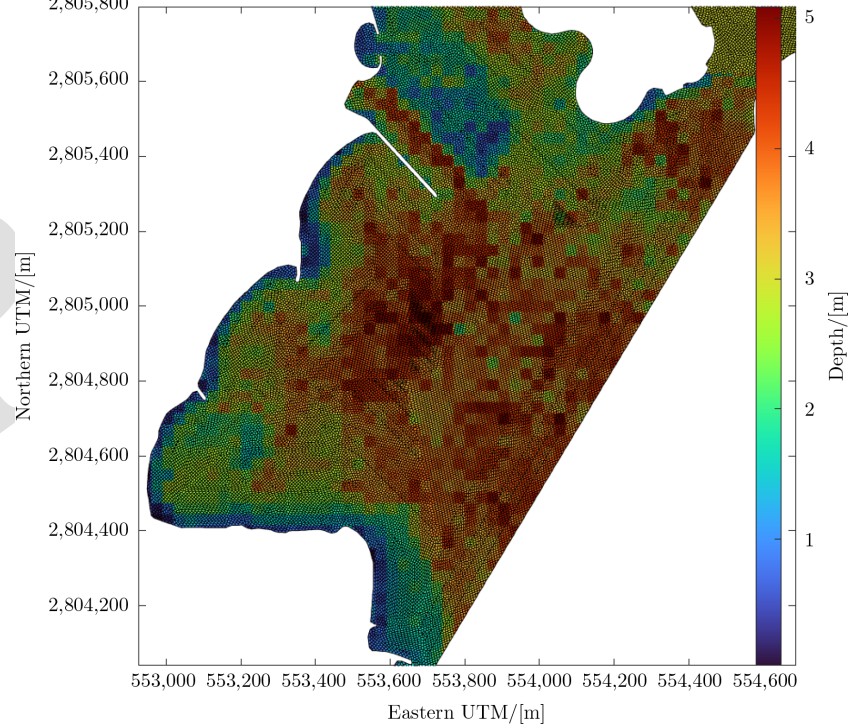

**Figure 6.** The wave-resolving high-resolution mesh to bring into focus the propagation of high-energy waves. The northern portion of the mesh is not shown to resolve individual cells in the plot.

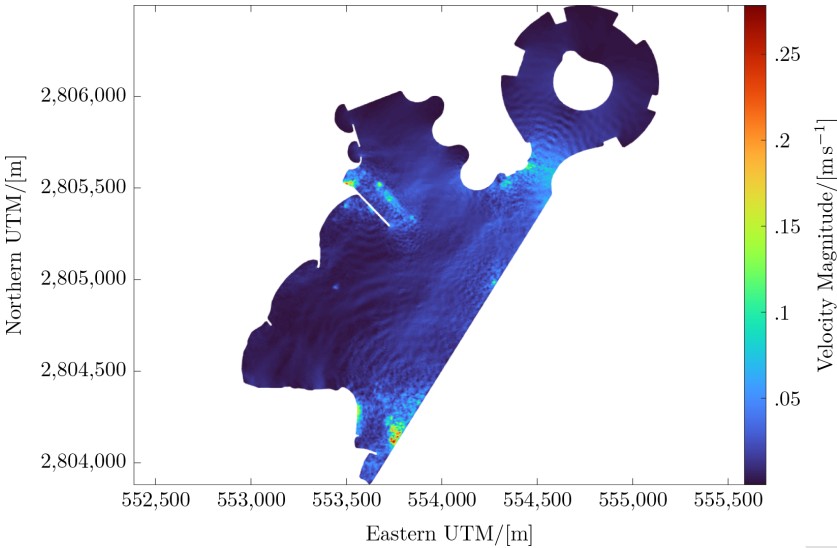

**Figure 7.** Wave-resolving simulation of the tidal regime, showing the velocity magnitude distribution.

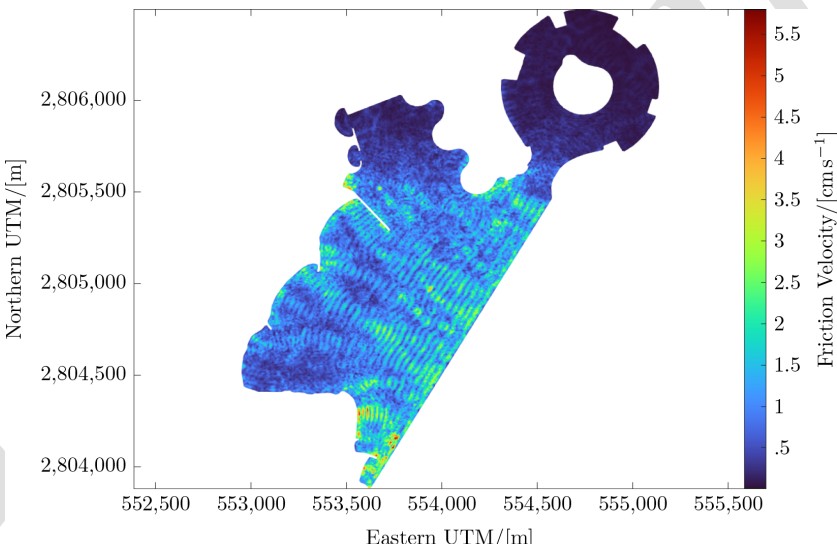

**Figure 8.** Wave-resolving simulation of the high-energy regime, showing a moment in time for the dynamic bottom-friction velocity distribution. The latter governs the erosion model in terms of the Rouse number.

observed processes: they show the potential for erosion at the southernmost beach and sediment settling at the sheltered beach in the southwest. Sediment settling was found to occur in the sheltered southernmost one of the three crescent beaches in both the model and observations. Erosion is impeded adjacent to groynes. That is, the wave-resolving simulation brought fine patterns into focus that might not be recovered without the high resolution that was used, enhancing coastal management. This encourages researchers to conduct sediment transport simulations on Voronoi-mesh-based platforms and with high resolution enabled by parallelization or GPU acceleration. The wave-resolving simulation (Fig. 8) resolves the wave-driven dynamics that dominate the friction velocity and wave attenuation in the marina north of the development.

## 4 Conclusions

Simulated surface elevations have been validated with five time series from three tidal meters and for two seasons, April and August 2023. The model exceeds real-time performance on a Ryzen 9 or comparable desktop CPU. Vertical current profile data were used to calibrate and conduct a sensitivity study for the roughness length, boundary conditions were set based on two tidal meters, and the validation was conducted

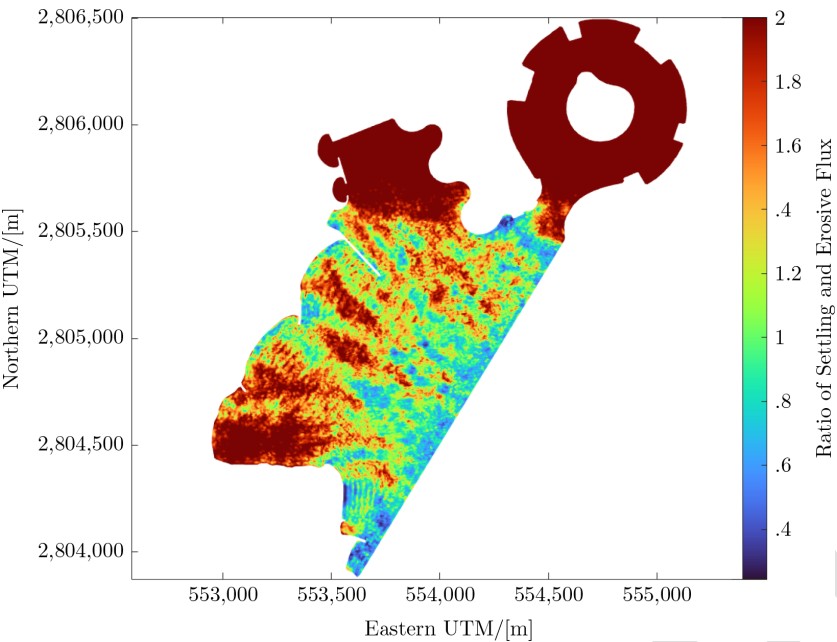

**Figure 9.** Ratio of sediment settling and erosion fluxes, showing sediment settling at the southeastern end of the beaches depicted (red), erosion (blue) at the southernmost beach, and the functioning of the groynes. The northern groyne that is tilted to the south shelters its southern side.

based on data from three locations, totaling five locations for validation, calibration, and boundary conditions.

The correlation between simulated and surveyed surface elevation time series exhibited an exceptionally precise cor-
relation, with data and simulation for some plots being visually identical, yielding high confidence in the model results. The mean error and RMSE were consistently below 7 %, as visualized in Figs. 4 and 5 and compiled in Table 3. With Voronoi-capable models, a reduction in cell count; numerical
diffusion (Holleman et al., 2013; Chan et al., 2018); and, as demonstrated here, acute polygon angles can all be achieved. The model structure aligns with seamless pre- and post-processing in Matlab (as documented in the paper), automatic parallelization, and seamless GPU acceleration. Docu-
mented are different approximations of some terms from the Navier–Stokes PDEs, which are listed in Table 1. These alternatives additionally provide cross-correlation between different solvers, readily providing a discrepancy-based error estimate for adaptive time stepping.

The model comes with a comprehensive environment of modules: a remote sensing module with spiking neuron filtration, published previously (Lawen et al., 2022); its pollutant fate transport model for nonlinear conversion, published previously (Lawen et al., 2013, 2014); and the Voronoi mesh
generator, published separately. The simulated, highly resolved dynamic Rouse number distribution, the ratio between sediment settling and erosive flux displayed in Fig. 9, accounts for orbital wave motion. Otherwise unresolved details in settling and erosion, particularly adjacent to the groynes,

have been recovered in Fig. 9. Wave-resolved simulations can, therefore, considerably enhance coastal management.

Voronoi schemes can be expanded to $n$ dimensions, which might not improve results for coastal systems: the usual approach (Lawen et al., 2010, 2013, 2014) to resolving the vertical via multiple layers retains an alignment with the dominant horizontal current components and, thus, avoids numerical diffusion. That is, retaining multiple layers achieves quasi-flow alignment for the vertical. This caution might not hold for modeling wave breaking or moving coastal meshes (4D Voronoi). For comprehensiveness, the development of a global model may be a subsequent stage, a step that likewise provides boundary conditions for regional and local models (Holleman and Stacey, 2014; Chou et al., 2015).

## Appendix A: Bathymetry survey coverage

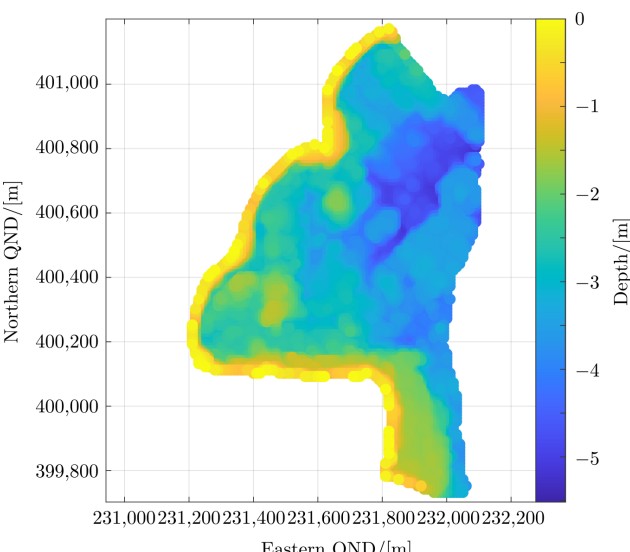

**Figure A1.** Bathymetry survey for modeled beaches. Domain sections outside the survey area were augmented with remote sensing. The remote sensing methodology has been published separately (Lawen et al., 2022).

*Code availability.* The Matlab version of the model can be accessed free of cost at https://www.environment.report/wavedyne.html (Lawen, 2025b). Bathymetry measurements to run the model for the relevant area can be obtained with the code provided at https://environment.report/spike-neuron-bathy.html (Lawen, 2025a).

*Data availability.* The Landsat satellite imagery used for remote sensing is publicly available through the United States Geological Survey (USGS) Earth Explorer platform at the URL https://earthexplorer.usgs.gov/ (United States Geological Survey, 2025).

*Competing interests.* The author has declared that there are no competing interests.

ther geographical representation in this paper. While Copernicus Publications makes every effort to include appropriate place names, the final responsibility lies with the authors.

*Special issue statement.* This article is part of the special issue "Oceanography at coastal scales: modelling, coupling, observations, and applications". It is not associated with a conference.

*Review statement.* This paper was edited by Davide Bonaldo and reviewed by two anonymous referees.

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

**Remarks from the language copy-editor**

**CE1**    The responses to CE2 and p. 8 line 42 have been updated. The other changes (including the response to CE3, which is more significant than simply reinserting 'with') involve more substantial edits to the accepted version of the paper and will need to be sent to the handling editor. To do this, please provide a detailed statement as to the rationale for why each change must be made. We must also have the explanations in order to send the proposed edits to the handling editor. For the statement, please provide a *.pdf file with the explanation which we can forward to the editor. Thank you for understanding.