# Peer review of "Wave-resolving Voronoi model of Rouse number for sediment entrainment"

_EGUsphere, 2024_

## Referee Comment (RC1)

**Review of "Wave-resolving Voronoi model of Rouse number for sediment entrainment equilibrium"**

Tuesday, June 11, 2024
4:48 PM

This is a difficult paper to review, as the topic of solving dynamic fluid mechanics models ala Navier-Stokes is considered almost intractable in some circles. Insight is not easy to come by when trying to understand a fully 3-dimensional space involving vortices, wave breaking, and turbulence. So I preface my remarks with this passage that Sir George Stokes wrote in response to reviewing Reynolds' paper on turbulent flow:

> *Dear Lord Rayleigh, I must plead guilty to not having digested Professor Osborne Reynolds's paper, though much time has passed since it was referred to me. I find it very difficult to make out what the author's notions are. As far as I can conjecture his meaning, I must say that I do not think that he has made out his point. He is however an able man, and in his former paper did very good work in showing that the condition of dynamic similarity which follow from the dimensions of the hydro-dynamical equations when viscosity is taken into account are not confined to what I may call regular motions, but continue to apply (in relation to mean effects) even when the motion is of that irregular kind which constituted eddies, and which at first sight appears to defy mathematical treatment. The fact that the author has gone to the expense of printing the paper shows that he himself considers it as of much importance. I confess I am not prepared to endorse that opinion myself, but neither can I say that it may not be true. I do not know if these remarks will be of any use in assisting the Council to come to a decision. Yours very truly, G.G. Stokes [1]*

Hope for progress will be in the ability to effectively model the empirical observations of wave behavior. To that extent the comparison of the measured sea-level heights in the location (Doha Bay) under evaluation against the model results are a vital aspect of the paper. Consider Figure 4 reproduced below. This appears to be a mixed diurnal-semidiurnal tidal time-series as the diurnal period is a shade under one day.

[Figure]

Figure 4: Correlation of simulated and surveyed surface elevation at the location of current meter 2 which also recorded depth, that is, surface elevation in August 2023.

Compare against a typical time-series from other locations such as the following figure [2], where the Manila and San Francisco both show mixed, with the Manila predominately diurnal and San Francisco more semi-diurnal.

[Figure]

As these time-series all have similar profiles for specific values of diurnal/semidiurnal mixing levels, it's not clear what the model resolving powers are of the paper's particular fit. In fact, most of the response is a direct response to the sinusoidal composition of the **K1** & **O1** diurnal tidal forcing and the **M2** & **S2** semidiurnal. It's a matter of whether the response was derived from first principles or by a training-validation calibration on an independent set of time-series data. From the text, it appears to be the latter.

> *"Two survey locations served to specify the boundary forcing and three survey locations served to validate the accuracy of the simulation. Given that two seasons have been examined, four time series were available for boundary forcing and six to validate the simulation. "*

As specifying boundary conditions is a euphemism for temporally aligning a forced response, it may be a good bet that the validated set will show a good correlation to a calibrated model. As Doha Bay is in the Persian Gulf, a mixed tidal response are reported as applicable, with the direct gravitational forcing of **K1**, **O1**, **M2**, **S2** applicable:

Or this from [3] showing again the mixed frequency mode of San Francisco

[Figure]

**Figure 1.2** (a) Tidal predictions for March 2043 at five sites that have very different tidal regimes. At Karumba, Australia, the tides are diurnal, at San Francisco, United States, they are mixed, whereas at both Mombasa, Kenya, and Bermuda, semidiurnal tides are dominant. The tides at Courtown, Ireland, are strongly distorted by the influence of the shallow waters of the Irish Sea.
(b) The lunar characteristics responsible for these tidal patterns. Solar and lunar tide-producing forces combine at new and full Moon to give large spring tidal ranges every 14.76 days. Lunar declination north and south of the equator varies over a 27.21-day period. Solar declination is zero on 21 March. Lunar distance varies through perigee and apogee over a 27.55-day period. The thumbnail cartoons show the physics of the variations (lunar phase, declination and distance) discussed in detail in Chapter 3.

> *The Persian Gulf, Figure 5.16, is a shallow sea with mixed diurnal and semidiurnal tides [29]. It is a largely enclosed basin with only a limited connection to the Indian Ocean through the Strait of Hormuz. Along the major northwest to southeast axis it has a length of about 850 km. The average depth is approximately 50 m, giving a resonant period near 21 hours, according to Equation 5.5. The Rossby radius at 27° N (c/f) is 335 km, comparable with the basin width. As a result the response to the diurnal forcing through the Strait of Hormuz is a single half-wave basin oscillation with an anticlockwise amphidrome. The semidiurnal tides develop two anticlockwise amphidromic systems, with*

*a node or anti-amphidrome in the middle of the basin. Near the centre of the basin the changes in tidal level are predominately semidiurnal, whereas near the semidiurnal amphidromes they are mainly diurnal. At the northwest and southeast ends of the basin the tidal levels have mixed diurnal and semidiurnal characteristics. **Direct gravitational forcing is probably significant.** [3]*

The concern here is that the direct gravitational forcing input is being reflected in the results as a flat or nearly equilibrium response. This is the first-order result of Laplace's tidal equations given a forcing input, and all the other details are aspects of a dynamic natural response or non-linear non-autonomous modulations of the forced response. For example, is there a possibility of detecting an amphidromic response of the Coriolis-effect in the equations to detect an anticlockwise cycle in the non-equilibrium dynamic s?  If not that at least provide a Fourier series spectrum of the sea elevation time-series so that any non-linear impacts of the model appear as harmonics or mixed harmonics of the direct forcing.  This will likely require a more densely sampled and longer time series.

[1] https://hal.science/hal-03378653/document
[2] Kvale, Erik P. The Origin of Neap-Spring Tidal Cycles, Marine Geology 235 (2006) 5 – 18
[3] Pugh, David., Woodworth, P. L.., Woodworth, Philip. Sea-Level Science: Understanding Tides, Surges, Tsunamis and Mean Sea-Level Changes. United Kingdom: Cambridge University Press, 2014.

---

## Referee Comment (RC2)

Review of "Wave-resolving Voronoi model of Rouse number for sediment entrainment equilibrium"

This is an interesting paper but it is not easy to review as there is a lot of condensed information.  There are some questions that arise from this work.

1. The author states in the introduction that Voronoi approximations exhibit a reduction in numerical diffusion vs Delaunay meshes.  The  work that the author refers [1], investigates the relationship between grid alignment and diffusive errors in the context of scalar transport in a triangular, unstructured, 3-D hydrodynamic code.  They conclude that the flow-aligned grids (for first order upwind advection) eliminate the lateral numerical diffusion. How is this related to the Voronoi meshes?

2. What is the cpu time that you need for the bay simulation of 8 days? What is happening, concerning he numerical diffusion, if you run for longer time?

3. The author states " Voronoi meshing has lately also been applied to oceanography with works [10, 11] mentioning different stability concerns vs. Delaunay meshes, indicating that an algorithm that might be stable on a Delaunay mesh might not necessarily be stable on a Voronoi mesh". How do you confirm the stability of your scheme on the Voronoi mesh?

4. What is the order of the scheme in space and time? What time integrator do you use? Is it possible to extend you scheme in higher dimensions? Does this have a meaning?

5. Does the solution of the sediment transport and settling happens is a coupled way in respect with the other equations? How do you account for the bed evolution?

6. Please correct reference [4]. The correct one is the [1] here.  The authors refer to Chapter 8 p. 597 of Randall and Bonny but we can not find it in references.

[1] Rusty Holleman, Oliver Fringer, and Mark Stacey. Numerical diffusion for flow-aligned unstructured grids with application to estuarine modeling. Int. J. Numer. Methods Fluids, 72(11):1117–1145, 2013.

---

## Author Response (AR1)

@Reviewer 1: Thank you for the review and part pertaining to Reynolds. I have added in the conclusion section that, to tackle large-scale tidal simulations, the development of a global model should be the next step. Added "For comprehensiveness, the development of a global model may be a subsequent stage, a step that likewise provides boundary conditions for regional and local models".

I have increased in Figure 5 the mark density and increased the duration by 30 to 40% to balance coverage and readability of the plot. If a further increase is advised, then I can go up to twenty-x days as per the available data.

[Figure]

Figure 5: Correlation of simulated and surveyed surface elevation at the location of tidal meter 2 (left) and 3 (right) during August (top) and April (bottom) of 2023.

@ Reviewer 2: Thank you for the review questions. Manuscript got revised with changes as per points 1, 2, 3, 4 and 6, as detailed below.

1. The reference to Holleman et al. (2013) alongside Chan et al. (2018) is made to point out the

significance of numerical diffusion. The connection to Voronoi meshes is drawn by the second citation in the bracket, i.e. Chan et al. (2018). Voronoi meshes exhibit fewer acute polygon angles (spanned by cell vertices that are acutely remote from the cell centroid) which factor into numerical diffusion. Of course, flow aligned meshes are even better to tackle numerical diffusion but are here impeded by tidal dynamics bar the mesh itself becoming dynamic. Have changed "Furthermore, Voronoi approximations exhibit a reduction in numerical diffusion vs Delaunay meshes [4, 5]" to "In terms of numerical diffusion [4], Voronoi meshes exhibit a reduction compared to Delaunay meshes [5]." in the pending Arxiv revision.

2. Absent GPU acceleration, running this simulation along 5 other hydrodynamic simulations took one week on a Ryzen 9 7950X3D. That translates to about one day if all the CPU cores are dedicated to the simulation. Have added "The model exceeds real-time performance on a Ryzen 9 or comparable desktop CPU".  Unboudedness over time is not a concern for numerical diffusion as it has a nivellating effect. In fact, its dampening effect can mitigate overshooting at the expense of accuracy.  Have added "numerical diffusion does not threaten the stability but accuracy of a simulation".

3. The algorithm has been validated with a method of manufactured solutions, MMS, which has been submitted separately for publication. The MMS was realized by oscillating the seabed to match the flow field to an analytical solution. The method can be used to validate the algorithm and mesh. Have included points mentioned in 3. and 4. below in the pending Arxiv revision. Have added "Algorithm validation has been conducted with a method of manufactured solutions (MMS), which has been submitted separately for publication \cite{Lawen24}. The MMS was realized by oscillating the seabed to match the flow field to an analytical solution." in the manuscript.

4. Added "The method is first order in space and time to attain high resolution meshes (Figure 6) to resolve waves while remaining efficient in terms of Flops: to resolve waves, the cell size should be a log order below the part of the wave spectrum of interest, i.e. maximizing cell count and minimizing Flops per cell."

Timewise the LHS in equations such as #11 denotes forward Euler approximation. The $^{+\delta t}$ denotes a quantity at the subsequent time level. Past Delaunay versions of a species transport model (Lawen et al., 2013, Lawen et al., 2014 as cited in the paper), which worked in conjunction with other ocean models, offered for scalar quantities also semi-implicit matrix reordering algos. However, these attained only a tripling of time steps at the expense of flops for the reordering, rendering the net gain in terms of flops questionable. Added in the MS "Past Delaunay versions of the 3D SMART's species transport \cite{LAWEN2013330, Lawen2014} offered for scalar quantities also semi-implicit matrix reordering algorithms. However, these attained only a tripling of time steps at the expense of Flops for the reordering, rendering the net compute gain questionable."

Based on this experience, a split approach is pursued: fast explicit Voronoi algorithm (small timesteps but less flops per step) and slow implicit reference solver (published separately) for algo cross-validation in addition to validation with MMS and survey.

Yes, Voronoi schemes can be expanded to n dimensions. Interestingly, in this context, that might not improve results: for example, in the coastal case, the usual approach of resolving the vertical rather via multiple layers retains an alignment with the dominant horizontal current components and, thus, avoids numerical diffusion. That is, utilizing multiple layers achieves quasi flow alignment for the vertical. This example harkens back to question 1 above about numerical diffusion. But perhaps expanding the Voronoi tessellation to the vertical could be an enhancement to model wave breaking and moving coastal meshes (4D Voronoi). Added to the conclusion section "Voronoi schemes can be expanded to n dimensions. For coastal systems that might not improve results: the usual approach \cite{} of resolving the vertical rather via multiple layers retains an alignment with the dominant horizontal current components and, thus, avoids numerical diffusion. That is, retaining multiple layers achieves quasi flow alignment for the vertical. This caution might not hold for modeling wave breaking or moving coastal meshes (4D Voronoi)."

5. The model is capable of back-coupling modeled quantities onto hydrodynamic properties. There are two ways to account for bed evolution: A.) Bed thickness change fluxes are modeled for representative periods, such as a neap-spring cycle, and then extrapolated for an annual seabed update. B.) To really simulate multiple years. The latter would preferably be supported by GPU acceleration.

6. Yes, did change "Numerical diffusion for flow-aligned unstructured grids" to "Numerical diffusion for flow-aligned unstructured grids with applications to estuarine modeling". Thank you.